# Understanding of the Mechanism for Laser Ablation-Assisted Patterning of Graphene/ITO Double Layers: Role of Effective Thermal Energy Transfer

**DOI:** 10.3390/mi11090821

**Published:** 2020-08-29

**Authors:** Hyung Seok Ryu, Hong-Seok Kim, Daeyoon Kim, Sang Jun Lee, Wonjoon Choi, Sang Jik Kwon, Jae-Hee Han, Eou-Sik Cho

**Affiliations:** 1Department of Electronic Engineering, Gachon University, Gyeonggi-do 13120, Korea; village1994@naver.com (H.S.R.); sjkwon@gachon.ac.kr (S.J.K.); 2Department of Materials Science and Engineering, Gachon University, Gyeonggi-do 13120, Korea; hskim2024@gmail.com; 3Department of Energy IT, Gachon University, Gyeonggi-do 13120, Korea; sismons@naver.com; 4School of Mechanical Engineering, Korea University, 145 Anam-ro, Seongbuk-gu, Seoul 02841, Korea; esj1016@korea.ac.kr (S.J.L.); wojchoi@korea.ac.kr (W.C.)

**Keywords:** graphene, ITO, laser ablation, thermal-energy transfer, temperature distribution, Raman spectroscopy

## Abstract

Demand for the fabrication of high-performance, transparent electronic devices with improved electronic and mechanical properties is significantly increasing for various applications. In this context, it is essential to develop highly transparent and conductive electrodes for the realization of such devices. To this end, in this work, a chemical vapor deposition (CVD)-grown graphene was transferred to both glass and polyethylene terephthalate (PET) substrates that had been pre-coated with an indium tin oxide (ITO) layer and then subsequently patterned by using a laser-ablation method for a low-cost, simple, and high-throughput process. A comparison of the results of the laser ablation of such a graphene/ITO double layer with those of the ITO single-layered films reveals that a larger amount of effective thermal energy of the laser used is transferred in the lateral direction along the graphene upper layer in the graphene/ITO double-layered structure, attributable to the high thermal conductivity of graphene. The transferred thermal energy is expected to melt and evaporate the lower ITO layer at a relatively lower threshold energy of laser ablation. The transient analysis of the temperature profiles indicates that the graphene layers can act as both an effective thermal diffuser and converter for the planar heat transfer. Raman spectroscopy was used to investigate the graphite peak on the ITO layer where the graphene upper layer was selectively removed because of the incomplete heating and removal process for the ITO layer by the laterally transferred effective thermal energy of the laser beam. Our approach could have broad implications for designing highly transparent and conductive electrodes as well as a new way of nanoscale patterning for other optoelectronic-device applications using laser-ablation methods.

## 1. Introduction

For the fabrication of transparent electronic devices, such as touchscreen panels (TSPs) and flat-panel displays (FPDs), it is essential to develop transparent conductive electrodes (TCEs) with high transmittance and low resistivity. Although indium tin oxide (ITO) has been mainly applied as a typical material for TCEs in electronic devices because of its high figure of merit and good compatibility with conventional semiconductor-processing technology [1,2,3], the demand for developing new TCE materials with better conductivity has been soaring as the sizes of TSPs and FPDs increase continuously [4,5,6,7,8,9,10,11].

Graphene has been widely attempted to be used as one of the promising candidates for TCEs in various optical and electronic device applications owing to its high transmittance and high flexibility [12,13,14]. These properties are known to stem from the extremely thin, one-carbon-atom thickness, uniform absorption of light causing a zero-energy bandgap, and high carrier mobility because of its two-dimensional (2D) sp^2^ electronic hybridization configuration [15,16,17]. On the other hand, it is difficult to apply graphene in industrial processes because of its relatively high sheet resistance as compensation for its extremely thin thickness and fragility. Therefore, it is necessary to solve the demerits of the graphene layer for applications such as TCEs.

In this study, to realize a higher conductance of a TCE in an electronic device, a chemical vapor deposition (CVD)-grown graphene layer was transferred to indium tin oxide (ITO)-film-coated glass or polyethylene terephthalate (PET) substrates, and graphene on an ITO (graphene/ITO) double layer was suggested as a new TCE material. However, it is very difficult to produce fine electrode patterns because of the difference in etch selectivity between graphene and ITO layers. In the case of the wet etching process for the removal of the ITO layer, it is indeed challenging to maintain graphene-electrode patterns without damage. The electron-beam-lithography and plasma-etching techniques have been reported as some patterning methods for graphene [18,19,20]. On the other hand, such conventional lithography methods would be undesirable for patterning graphene/ITO because of their highly costly vacuum process. Therefore, for low-cost patterning with high throughput, we demonstrate a direct laser ablation method to pattern the graphene/ITO double-layered films on glass and PET substrates. In the fabrication of TSPs, laser ablation has been applied to various transparent oxides including graphene without a lithography mask [21,22,23,24,25,26]. As a method of relatively low-cost patterning, a pulsed laser with a wavelength of 1064 nm was used for direct laser patterning under various laser-beam conditions, and the experimental results were analyzed in terms of the laser ablation threshold of the graphene/ITO double layer for the optimization of the laser-patterning process.

## 2. Materials and Methods

A-few layered graphene films were produced in a CVD furnace and transferred to soda-lime-glass and PET substrates [27,28,29], on which a 70 Å-thick ITO layer had been deposited. To fabricate either graphene/ITO/glass or graphene/ITO/PET substrates as a platform for the laser ablation studied in this work, the graphene layers were first synthesized as follows. After the copper (Cu) foil as a substrate for synthesis of graphene layers was heated in a furnace up to 1050 °C for 60 min in hydrogen (H_2_) at a flow rate of 60 sccm, a few-layered graphene film was grown with methane (CH_4_) feedstock gas at a flow rate of 5 sccm. Subsequently, polymethyl methacrylate (PMMA) was spin-coated over the CVD-grown graphene layer on the Cu foil; then, the whole PMMA/graphene/Cu foil structure with Cu foil downside was carefully floated on ammonium persulfate (APS, (NH_4_)_2_S_2_O_8,_ Sigma-Aldrich Korea Ltd., Seoul, Korea)) solution to etch off the Cu foil [30,31], followed by rinsing the remainder using deionized water. Then, the remaining PMMA/graphene structure, still floating, was transferred to the target substrates, which were the ITO-film-coated glass or PET substrates. Finally, only the uppermost PMMA layer was immediately removed with acetone, resulting in the final structures of either the graphene/ITO/glass or graphene/ITO/PET substrates. Appendix A in Appendix A lists the optical transmittance and sheet resistance of the ITO single and graphene/ITO double layers on glass and PET substrates. The transmittance with wavelengths ranging from 400 to 800 nm and sheet resistance were measured using a UV-visible spectrometer (Cary 100 UV-Vis, Agilent Technologies, Inc., Santa Clara, CA, USA) and 4-point probe (CMT-SR2000N, AIT Co., Ltd., Suwon, Gyeonggi-do, Korea), respectively. Compared to the ITO/PET substrate without a graphene layer, the 1-layer-graphene/ITO/PET or even the 2-layer-graphene/ITO/PET substrate showed similar transmittance but a trend towards a decrease in the sheet resistance with an increase in the number of graphene layers [32], despite the contact resistance between the ITO and graphene applied (Appendix A). These results confirm that the sheet resistances obtained were appropriately measured in this work. In addition, the data for the transmittance for different ITO-coated substrates with and without a graphene layer are also included in Appendix A.

A 1064 nm, Q-switched, diode-pumped, neodymium-doped yttrium vanadate (Nd:YVO_4_) laser system (Miyachi, ML-7111A, Miyachi Korea Co., Sungnam, Gyeonggi-do, Korea) was used for the direct patterning of the prepared graphene/ITO-layered films (Figure 1). The pulse duration (τ) of the laser beam was 10 ns for a full width at half maximum (FWHM), and the repetition rate ranged from 0 to 200 kHz. The samples were laser ablated under various laser-beam conditions, such as different repetition rates and scanning speeds. During laser ablation, the beam energy per laser pulse was adjusted from 44.6 to 266.5 μJ according to the repetition rate.

## 3. Results and Discussion

Figure 2 shows the relationship between the squared laser-ablated spot size on the thin film and the laser-beam fluence per etch pulse to determine the ablation threshold value of each layer of the graphene/ITO-double-layer structure [25,33,34,35,36,37,38]. Using the equation $D^{2}=2\omega_{0}^{2}\;\times \ln\left(\frac{E_{0}}{E_{th}}\right)$, from the ablated spot size data *D* per laser energy pulse *E*_0_, the laser beam radius ω_0_ can be obtained from the slope of the linear-fitted graph. Considering the equation $\Phi =\frac{2\times E}{\pi \times \omega_{0}^{2}}$, the ablation threshold values can be obtained by extrapolating the fitting lines to the intercepts of the logarithmic axis of the laser-beam fluence. From Figure 2a, the ablation threshold values of the laser-beam fluence obtained for both the graphene upper and ITO lower layers of the graphene/ITO double-layer structure on glass substrates were 7.54 and 0.86 J/cm^2^, respectively. An accurate value of the ablation threshold for the ITO single layer on glass could not be determined because the ablation threshold for the ITO layer increases when its thickness is much smaller than its absorption length [37]. Most of the laser beam is expected to transmit through both the thin ITO and glass substrate instead of being absorbed due to the relatively long wavelength of 1064 nm employed in this work. Therefore, the laser ablation process in this experiment is expected to be influenced mainly by the thermal energy of the laser beam. Furthermore, we also investigated the ablation threshold values for different layers on the PET substrates: the ITO single layer and both the graphene upper and ITO lower layers of the graphene/ITO double-layer structure on each PET substrate have the values of 1.50, 0.30, and 1.31 J/cm^2^, respectively (Figure 2b). Compared to the ablation threshold values with the glass substrate (Figure 2a), the lower values for each layer on the PET substrates can be explained as a result of the lower thermal conductivity of PET than glass substrate [39,40].

Optical images of the laser-ablated lines on the graphene/ITO double-layer structure on the glass substrate with a laser scanning rate of 1000 mm/s at different pulse energies are shown in Figure 3. When the laser pulse energy was 266.5 μJ, both the graphene upper and ITO lower layers were found to be etched to form stripes and spots, respectively, as shown in Figure 3a. In the case of 156.5 μJ, only small, etched spots on the ITO lower layer were found (Figure 3b), and furthermore, no ITO layer appeared to be laser ablated at lower pulsed energy, 88.1 and 44.6 μJ (Figure 3c,d). Although the laser scanning rate was increased to 2000 mm/s, a similar tendency to the result of Figure 3 was observed for the graphene/ITO double-layer structure on the PET substrate (Figure 4). The largest etched spots (~100 μm in diameter) were found with the pulsed energy of 266.5 μJ. As the pulsed energy was decreased from 266.5 to 88.1 μJ, the etched spots, even on the ITO lower layer, were still observed, and no spots were found in the case of 44.6 μJ. The etched spot pattern of the graphene upper layer in Figure 4a was attributed to the different overlapping rates as a result of the scanning speed [22,40,41,42]. The overlapping etched spot patterns of the graphene upper layer in Figure 3a and Figure 4b could be due to the overlapping of the laser beam.

The difference in the laser-driven etched spot patterns between the graphene upper and ITO lower layers can be explained as a result of the higher thermal conductivity of graphene than that of the ITO layer [43]. Therefore, it is possible to examine the larger etched spots on the graphene layer and the higher overlapping as the formation of stripes [40]. Compared to the ITO single layer, a lower ablation threshold and a larger etched spot were investigated on the ITO lower layer of the graphene/ITO double-layer structure. In addition, we calculated the lateral (or in-plane directional) etch rate for the different samples from our experimental results (Appendix A). The average values with standard deviation σ of the lateral (or in-plane directional) etch rates in unit of mm^2^/s for the ITO and graphene layers of the graphene/ITO/glass substrates were found to be 0.02781 (σ, 0.0064) and 0.27637 (σ, 0.00905), respectively. Those of the graphene/ITO/PET substrate were 0.18134 (σ, 0.1035) and 0.56833 (σ, 0.2951), respectively. There is a tendency that in the same substrate, the lateral etch rate of graphene is higher than that of the ITO layer. This could be because of the difference in the thermal conductivity as well as film thickness between the two layers. For the same material, for example, the graphene layer with PET shows a higher value with increasing σ compared to that with a glass substrate. This result would be possibly due to the lower thermal conductivity [39,40] as well as the possibility of a less-uniform distribution of the heat developed during the laser-ablation process for the former over the latter. Figure 5a,b show the expected effective thermal-energy transfer of the laser beam through the ITO single layer and graphene/ITO double-layer structure during the laser-ablation process, respectively. When a laser beam is incident on the graphene layer with a Gaussian distribution, its effective thermal energy is expected to be transferred in the lateral direction through the graphene upper layer [44]. The thermal conduction from the graphene upper layer to the ITO lower one can be explained using a heat-flow equation in one-dimensional form for the time-dependent temperature distribution in the laser-ablated graphene/ITO double-layered films [45]. The boundary condition at the interface between the graphene upper and ITO lower layers can be expressed for the heat flow in the equation $\kappa_{graphene}{\frac{\partial T}{\partial z}|}_{graphene}=\kappa_{ITO}{\frac{\partial T}{\partial z}|}_{ITO}$ [46]. κ is the thermal conductivity, and z is the distance into the vertical direction from the surface of the thin film to the substrate. From the above boundary condition, the conveyed effective thermal energy to the ITO lower layer of the graphene/ITO double-layer structure is expected to cause an abrupt change in temperature and ablate the ITO lower layer more easily in a larger area than the ITO single layer because of the difference in the thermal conductivity of graphene and ITO, as shown in Figure 5b.

The transient temperature distribution confirms the different characteristics of thermal-energy transport on graphene/ITO double and ITO single layers, enabled by the laser irradiation for 10 ns (Figure 6). The analysis of the temperature changes was conducted using Comsol Multiphysics [47]. First, the graphene/ITO double layers show effective thermal-energy transport, thereby facilitating heat dissipation on a two-dimensional plane, while the laser beam is irradiated for only 10 ns, as shown in Figure 6a. The transient changes in the temperature profiles for every 2 ns along the white-dotted line more clearly present the extended thermal function of the graphene layer on ITO film-coated substrates (Figure 6b). As the graphene/ITO double layer is exposed in the laser irradiation, it significantly absorbs the produced thermal energy, and the instant heat diffusion in the lateral direction appears on the x–y plane. The higher thermal conductivity of the graphene layer could significantly enhance the planar thermal-energy transport, despite the low thermal conductivity of the ITO layer. On the other hand, the thickness of the ITO layer film (~70 Å) is over 20 times greater than that of the graphene layer (~3 Å). While the thermal diffusion is guided in the planar direction of the graphene layer, the penetration of thermal energy passing through the vertical direction is minimized, and the desirable thermal path via the graphene layer facilitates highly anisotropic thermal diffusion. The different trend of temperature changes in the single ITO layer enabled by the laser irradiation clarify the distinct functions of the graphene layer in terms of the enhanced planar thermal diffusion. The laser-beam irradiation for 10 ns on the ITO-layer film forms much narrower thermal diffusion (~30 μm) along the planar direction than that on the graphene/ITO double layers (Figure 6c). The thermal diffusion length with the graphene layer (~70 μm) enhances the thermal diffusion length by more than two times. This trend is observed in the transient temperature changes for every 2 ns, as well (Figure 6d). Although the laser-beam irradiation increases the peak temperature at the central area for both cases, the effective thermal diffusion in the planar direction only increases in the graphene/ITO double layer. Furthermore, the maximum temperature on the graphene/ITO double layer reaches over 5000 K, whereas the ITO-single-layered film only shows about 1500 K. The transient analysis of the temperature profiles indicates that the graphene layers can act as both an effective thermal diffuser and converter for the planar heat transfer. In addition, despite the relatively low melting temperature (523 K [48] to 533 K [49]) of the PET substrate compared to the peak temperature (>1500 K) resulting from the simulation, it is reasonably expected that the PET substrate would not be completely melted with such a short laser-ablation duration (10 ns applied for both the experiment and simulation) but rather be cracked to release the stress developed during the process [39].

To confirm the removal of the graphene/ITO double layer after laser ablation, Raman spectroscopy was performed to examine a laser-etched spot on the graphene/ITO double layer (Figure 7a), where the regions 1, 2, and 3 were assigned to indicate the graphene upper layer, ITO lower layer, and PET substrate in sequence. On each part of the laser-ablated graphene patterns, the Raman spectra were compared, as shown in Figure 7b, from 200 to 3150 cm^−1^. For the region 1, the three characteristic bands, D-, G-, and 2D-band, indicative of the existence of a graphene layer were found [50,51,52,53,54,55,56,57]. A D-band peak (~1350 cm^−1^) of weak intensity and both G- (~1580 cm^−1^) and 2D-band peaks (~2700 cm^−1^) on the graphene layer were observed. When the Raman spectra were acquired along the edge of a laser-etched spot, where the graphene upper layer was etched selectively but the ITO lower layer remained (region 2), very small traces of both the D- and G-band peaks were observed, but no 2D-band peak was detected, indicating that most of the graphene upper layer had been removed as a result of laser ablation. Compared to the Raman spectra of a center region of the laser-etched spot, on which no graphene-related peaks were observed (region 3), some of the residual amorphous carbon on the ITO lower layer was found, which is expected to have resulted from the incomplete heating and removal process for the ITO lower layer by the laterally transferred effective thermal energy of the laser beam.

## 4. Conclusions

In conclusion, we experimentally showed the laser-ablation-assisted patterning of graphene/ITO double layers for future TCE applications. To elucidate the underlying mechanism for the difference between ITO single and graphene/ITO double layers during the laser-ablation process, we also conducted a theoretical analysis of transient temperature spatial distribution, revealing thermal-energy transport on graphene/ITO double and ITO single layers. From the analysis, we demonstrated a role of effective thermal-energy transfer during laser ablation, indicating that the graphene layers can act as both an effective thermal diffuser and converter for the planar heat transfer. The analysis of Raman spectroscopy also supported this laser-ablation mechanism of the selective etching of graphene layers due to the effective thermal-energy transfer. Our study may provide insight into designing optimal electrode patterning using graphene layers, which would not only enable better electrode formation with high transparency and conductance for TCE applications but would also serve as a new way of nanoscale patterning for other optoelectronic-device applications using laser-ablation methods. Future work will be carried out towards the further reduction of the size of the pattern width as well as the development of complex shapes of pattern design.

## Figures and Tables

**Figure 1 micromachines-11-00821-f001:**
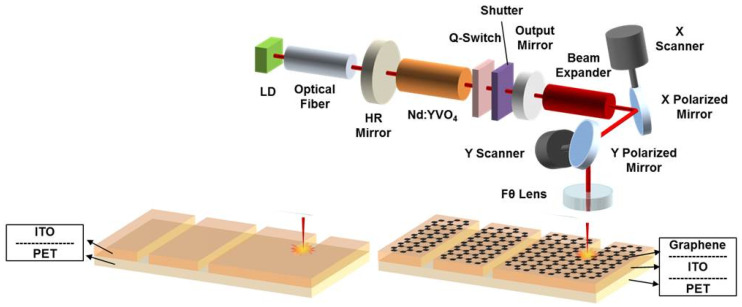
Schematic of the experimental setup for laser ablation of both indium tin oxide (ITO) single and graphene/ITO double layers on polyethylene terephthalate (PET) substrates.

**Figure 2 micromachines-11-00821-f002:**
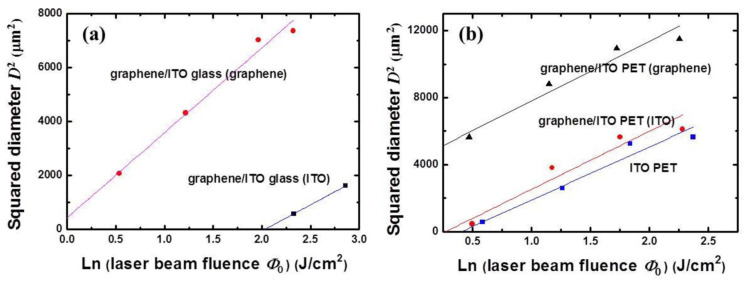
Squared laser-patterned diameters (line widths) of ITO single layer and graphene/ITO double layer on (**a**) glass, where the solid red circles and black squares indicate the data for graphene and ITO layers on ITO-coated glass substrate, respectively, and (**b**) PET substrates, where the solid black triangles and red circles indicate the data for graphene and ITO layers on ITO-coated PET substrate, respectively, the blue squares are the ITO single layer on PET substrate. Each colored line shows a linear fit of the data. The laser-ablated line widths of Figure 2a,b were obtained from Figure 3 and Figure 4, respectively.

**Figure 3 micromachines-11-00821-f003:**
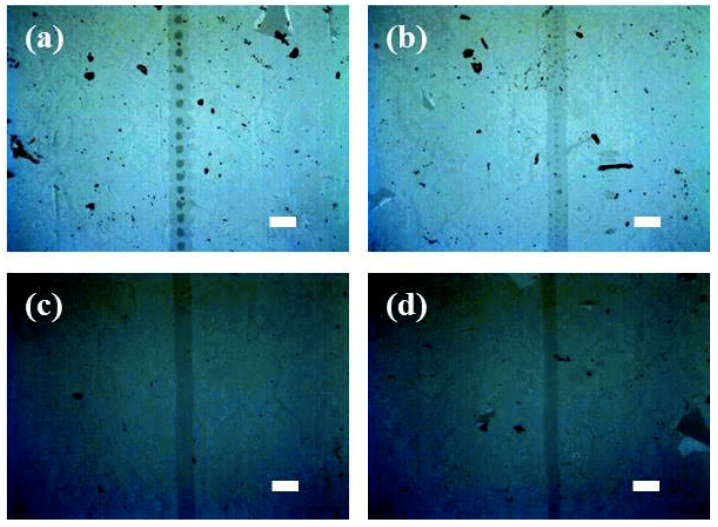
Optical microscopy images of the laser-patterned lines on the graphene/ITO double layer on a glass substrate produced with a scanning speed of 1000 mm/s at pulse energies of (**a**) 266.5, (**b**) 156.5, (**c**) 88.1, and (**d**) 44.6 μJ. White lines correspond to 100 μm.

**Figure 4 micromachines-11-00821-f004:**
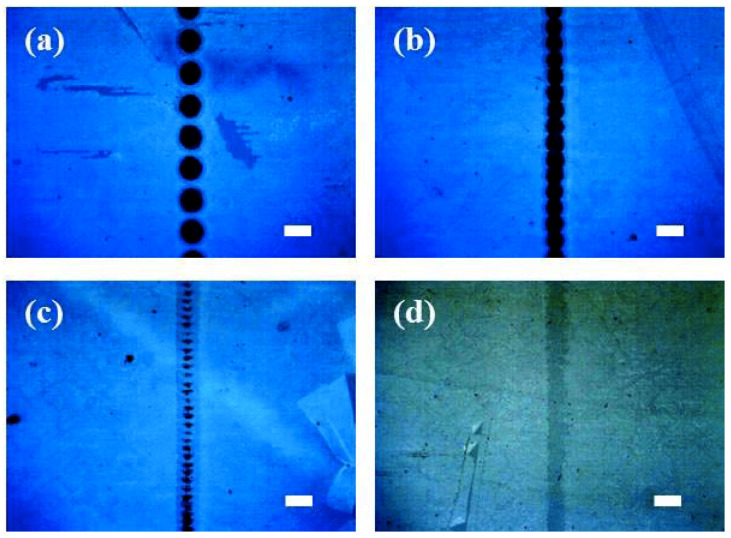
Optical microscopy images of the laser-patterned lines on graphene/ITO double layer on a PET substrate produced at a scanning speed of 2000 mm/s and pulse energies of (**a**) 266.5, (**b**) 156.5, (**c**) 88.1, and (**d**) 44.6 μJ. The white lines correspond to 100 μm.

**Figure 5 micromachines-11-00821-f005:**
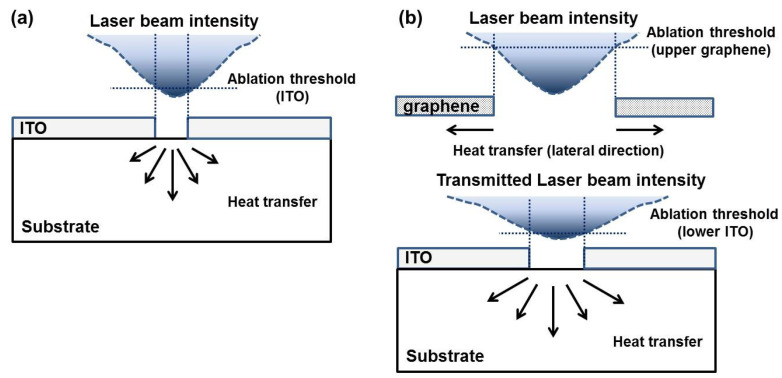
Schematic diagram of the thermal energy transfer of laser beam through (**a**) ITO single layer and (**b**) graphene/ITO double layer during laser ablation.

**Figure 6 micromachines-11-00821-f006:**
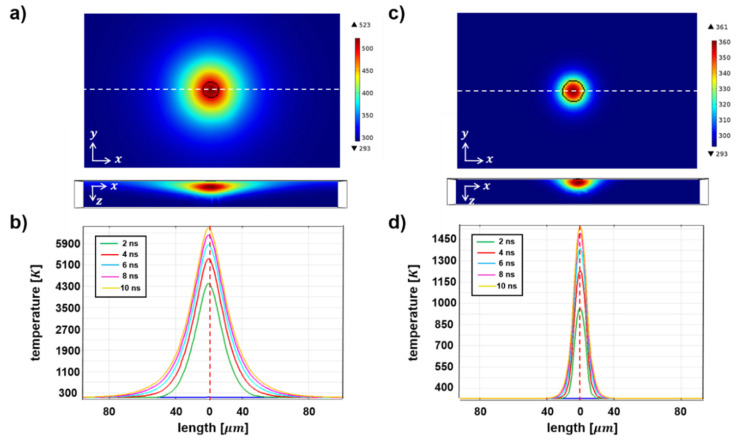
Transient analysis of temperature distribution on (**a**,**b**) graphene/ITO double layers and (**c**,**d**) ITO single layers, enabled by laser-beam irradiation. The two-dimensional features of instant temperature profiles with heating by laser-beam irradiation for 10 ns are shown in the x–y and x–z planes (**top**), and the corresponding transient temperature changes along the white dotted line are presented for every 2 ns (**bottom**).

**Figure 7 micromachines-11-00821-f007:**
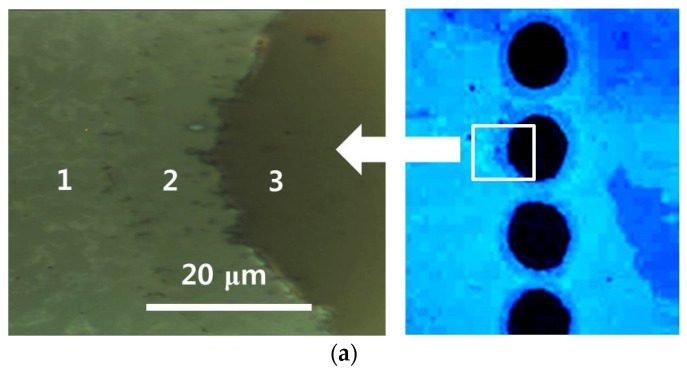
(**a**) Expanded optical microscopy images of Figure 4a. The graphene/ITO double layer was laser ablated at a pulse energy of 266.5 μJ with a scanning speed of 2000 mm/s; (**b**) Raman spectra on regions 1, 2, and 3 of (**a**).

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
