# Peer review of "Understanding of the Mechanism for Laser Ablation-Assisted Patterning of Graphene/ITO Double Layers: Role of Effective Thermal Energy Transfer"

_micromachines, 2020, doi:10.3390/mi11090821_

Round 1

Reviewer 1 Report

Authors present their research on using graphene/ITO double layer as transparent conducting oxide for display applications. They have grown graphene using CVD and then transferred to ITO coated glass with a thin layer of PMMA as the joint layer.  They have also presented laser ablation assisted patterning of the graphene/ITO double layers along with thermal transfer.
The paper has good content and can be accepted pending the revision.

1) Page 2, line 75: PMMA was coated on graphene and then transferred to ITO. Hence it should be graphene/PMMA/ITO. Hence I am not convinced how Table S1 would be correct. It is not graphene/ITO junction as authors have mentioned. Please check the geometry and measurements. This part should be clear before accepting this paper.

2) Another aspect of Table S1 is that how authors have conducted the measurements. If the measurements were not carried out at multiple locations using a suitable probe station, the probe will penetrate the graphene and will be measuring the ITO conductivity. Authors may have to comment this aspect in the revised manuscript.

3) Figure 1: The cross-section presented is Graphene/ITO/PET. However, PMMA layer is missing.
I believe it is Graphe/PMMA/ITO/PET

4) Figure 2 caption is not clear. Please include (a) and (b) with more explanation

5) Page 7, Line 75: How did authors simulate 3 A thick graphene on COMSOL? Please elaborate simulation geometry and details?

6) Page 2: line 45: Authors may include other conducting oxide work using TiO2 in the reference such as 
Qing Dai, R.Rajasekharan, Haider Butt, Xiaohui Qiu, Gehan Amaragtunga, Timothy D Wilkinson, Ultrasmall microlens array based on vertically aligned carbon nanofibers,
Small 8 (16), 2501-2504 (2012)

Taro Hitosugi et a, Properties of TiO2‐based transparent conducting oxides, https://doi.org/10.1002/pssa.200983774

Reviewer 2 Report

No comments

Reviewer 3 Report

The authors are investigating transparent conductive electrodes by graphene and ITO. But it lacks some critical information. So major revision is necessary before considering for publication.

  1. After etching by using laser ablation assisted method, is there any transmittance change of PET layer? Experiment data required.
  2. What is the etching efficiency (mm2/s) for ITO or graphene, respectively?
  3. The simulation results showed peak temperature >1000K, what is the melting temperature for PET substrate? Would this simulation results be problematic or the experiment results for after etch is not from PET substrate?
  4. When the simulation with COMSOL is performed, was the PET substrate considered?

Round 2

Reviewer 3 Report

Thanks for the authors to response the technical questions from the reviewer. I would suggest accept as is.